EMBO
Molecular Medicine

# CAR T-cell-associated secondary malignancies challenge current pharmacovigilance concepts

Philipp Berg [ID][1], Gabriele Ruppert-Seipp[1], Susanne Müller[1], Gabriele D Maurer [ID][1], Jessica Hartmann [ID][2], Udo Holtick[3], Christian J Buchholz [ID][2][✉] & Markus B Funk [ID][1][✉]

## Abstract

**Suspected adverse reactions following chimeric antigen receptor T-cell (CAR T) treatment include more and more cases of secondary T-cell malignancies. The causality assessment of such suspected reactions challenges established evaluation practices due to (i) patient and product-specific risk factors and (ii) incomplete data available with post-marketing reports submitted to competent authorities. This is of particular relevance for gene therapy products that integrate into the host genome. We present a summary of case reports related to different CAR T products and the rationale for case causality assessment. In this context, possible pathophysiologic mechanisms and differences between CAR T products to be taken into account are discussed. The unparalleled complexity of the case follow-up and the multistep process of cancer development necessitates a case-by-case consideration. This highlights challenges in the pharmacovigilance of advanced therapy medicinal products and underlines the importance of testing for vector presence, integration location and gene expression profile for an informed case assessment of suspected secondary malignancies with the aim to obtain a better understanding of contributing factors.**

**Keywords** Gene Therapy; Insertional Mutagenesis; Oncogenesis; Adverse Reaction; Medicinal Safety

## Introduction

Chimeric antigen receptor T-cell (CAR T) therapies have become a promising treatment option for several haematological malignancies often resulting in durable remissions for patients with therapy-refractory diseases (Cappell and Kochenderfer, 2023). All currently approved CAR T products in the European Union (EU) consist of genetically modified autologous T cells that are transduced ex vivo using integrating, non-replicating lentiviral (LV) or γ-retroviral (γRV) vectors to express either an anti-CD19 or anti-B-cell maturation antigen (BCMA) CAR. The integration of genetic sequences into the host genome can potentially lead to insertional mutagenesis, which may affect the expression of genes directly or indirectly linked to cell proliferation, differentiation or survival, thereby promoting malignant transformation. This can occur through several mechanisms (Bushman, 2020). The risk of insertional oncogenesis was substantiated in hematopoietic stem cell gene therapy trials for X-linked severe combined immunodeficiency (SCID-X1) and Wiskott-Aldrich syndrome (WAS), where several patients developed acute T-cell leukaemia due to vector-mediated oncogene activation (Braun et al, 2014; Howe et al, 2008; Hacein-Bey-Abina et al, 2003). Enhancer elements present in first-generation γRV vectors used in these trials can alter the expression of adjacent genes, even if they are located several kilobase pairs away, thereby contributing to malignant transformation (Bushman, 2020). In newer generation of vectors this risk is significantly reduced by incorporating a vector modification known as self-inactivating (SIN) design (Zufferey et al, 1998). Despite these improvements, vector-mediated gene disruption can still occur, typically resulting in minor issues. Such disruptions are expected to cause primarily monoallelic, recessive defects and functional gene alterations are likely to lead to cell extinction rather than expansion due to a non-permissive genetic environment (Baum et al, 2004). Although vector safety has improved and no cases of leukaemia caused by vector integration have been reported since then, insertional mutagenesis leading to oncogenesis remains a safety concern for all vector-based gene therapies (CAT, 2013). Aside from one fatal case of unintentional transduction of a leukaemic B-cell during T-cell manufacturing (Ruella et al, 2018), which was unrelated to insertional oncogenesis, no CAR T-associated secondary malignancies have been reported for approved CAR T products until recently (Bonifant et al, 2016; Cappell and Kochenderfer, 2023). In late 2023 and early 2024, case reports of secondary malignancies of T-cell origin after CAR T therapy prompted investigations by the United States (US) Food and Drug Administration (FDA) and the European Medicines Agency (EMA). Following these announcements several articles were published by academic institutions that analysed publicly available data, contributed to the analysis of individual T-cell lymphoma cases, and raised questions regarding risk factors in the treated populations and the frequency and causality of such events (Lamble et al, 2024; Levine et al, 2024; Elsallab et al, 2024; Ghilardi et al, 2024; Hamilton et al, 2024). This is particularly important since CAR T-cell treatments are currently the most widely administered gene therapies, and the increasing number of patients treated raises the likelihood of detecting rare events. With over 40,000 patients having received approved CAR T

---

[1]Safety of Biomedicines and Diagnostics, Paul-Ehrlich-Institut, Langen, Germany. [2]Molecular Biotechnology and Gene Therapy, Paul-Ehrlich-Institut, Langen, Germany. [3]Department I of Internal Medicine, Medical Faculty and University Hospital Cologne, University of Cologne, Cologne, Germany. [✉]E-mail: Christian.Buchholz@pei.de; Markus.Funk@pei.de
https://doi.org/10.1038/s44321-024-00183-2 | Published online: 6 January 2025

products worldwide and an average of ~$10^8$ CAR-positive cells per patient, it is estimated that more than $10^{13}$ random vector insertions have occurred, especially since each cell can harbour multiple vector insertions.

Monitoring the safety of medicinal products following marketing approval is crucial to ensure a positive benefit-risk ratio throughout the life cycle of a drug. However, spontaneous reporting systems are limited by underreporting of adverse drug reactions (ADRs) (Hazell and Shakir, 2006) as well as varying quality and completeness of ADR post-marketing reports (Berg et al, 2023), meaning that the level of available data lags far behind those of a clinical trial. While the analysis of individual malignancy cases can and should be as thorough as possible (Harrison et al, 2023; Ghilardi et al, 2024), the assessment of pharmacovigilance reports of a heterogeneous nature must take these limitations into account.

In this article, we comment on the process of causality assessment of T-cell malignancy cases reported to the Paul-Ehrlich-Institut (PEI) following CAR T therapy and based on the experience from this evaluation, highlight the need for more adequate sample testing to aid evidence-based assessments.

## Modified criteria for causality assessment

Pharmacovigilance reports of T-cell malignancies after CAR T therapy require a standardised approach to evaluate a potential causal relationship in each case. Current practice follows established criteria that are widely used to aid the causality assessment of suspected adverse drug reactions (ADR) with the aim to determine the level of plausibility of the relationship between the ADR and a medicinal product (WHO-UMC system) (WHO, 2013). Since these criteria do not account for particularities of gene therapy products (e.g., one-time treatment, permanent vector insertion), we suggest to apply modified criteria to pharmacovigilance reports. In addition, it is noted that all patients receiving approved CAR T are per se at an increased risk for developing new malignancies due to potential confounders such as underlying disease, genetic predispositions, and previous lines of treatment. Thus, a secondary malignancy may be explained by such circumstances. Nevertheless, the presence of risk factors

does not preclude the possibility of a causal relationship with the treatment. This is particularly relevant in view of the mechanism of insertional mutagenesis. Genetic testing of tumour samples can answer whether tumour cells were transduced (i.e., contain the therapeutic gene) and where precisely the therapeutic gene had integrated in the genome. The expression of genes directly affected or adjacent to genomic insertions can be investigated to inform about implications on the level of transcription or translation.

In order to develop criteria that focus on the most decisive features, we suggest considering different scenarios with regard to a causal association of a new T-cell malignancy diagnosis following CAR T therapy and the risk of insertional mutagenesis:

1. Development of a T-cell malignancy independent of T-cell transduction during CAR T manufacturing: i.e. no vector detected in tumour cells or a mixed population of cells with and without the vector;
2. Transduction of a pre-existing tumour cell: e.g., presence of a not-yet diagnosed neoplasia of T-cell origin; mixed population of tumour cells with and without vector; vector insertion does not necessarily exert an advantage to the malignant cell;
3. T-cell lymphoma development due to insertional mutagenesis: i.e. therapeutic gene present in tumour cells; vector insertion contributing to the malignant transformation and phenotype, e.g., by affecting cellular proliferation, survival, and/or genomic instability; tumour cells consist of a primarily monoclonal cell population in terms of vector insertion site;
4. T-cell lymphoma development from a single transduced cell by a mechanism independent of vector insertion: i.e. therapeutic gene present in tumour cells but as a non-functional passenger event without any impact on cell fitness.

Based on these considerations, we suggest that the causality assessment of CAR T therapy should follow modified criteria with molecular analysis of a tumour sample as central element (Fig. 1). In brief, if genetic testing of a tumour sample demonstrated that it does not consist of transduced

(vector containing) cells, insertional mutagenesis is considered "unlikely". In the event of a newly diagnosed T-cell malignancy after CAR T therapy without any T-cell tumour material tested, it is assumed that vector integration can neither be confirmed nor excluded. Under those circumstances the neoplasm could have been caused by insertional mutagenesis or developed independently, e.g., because of prolonged immunosuppression or other predisposing factors. Thus, a causality classification of "possible" is derived. In case T-cell tumour material is tested and shown to consist of a vector containing clone, in particular if the genomic insertion(s) occurred in regions that could affect cell proliferation or fate, it is considered likely that this event has contributed to the malignant transformation ("probable"). Alternative risk factors such as predisposing germline mutations may play a role and affect the susceptibility of an individual. However, such factors were pre-existing and cannot fully explain the timing of a new malignancy and the presence of the therapeutic gene. For classification as "certain" additional evidence of relevant downstream effects of vector insertion(s) is required. If sample testing is ongoing, a case is assessed as "conditional". When the diagnosis was not medically confirmed or information on the temporal relationship was lacking, a case is considered "unassessable". The categories do not proof or refute a causal link but intend to classify the likelihood of a relationship between an ADR and a product in each case. Further adjustment of an assessment strategy may be necessary in future. The application of these modified criteria is exemplified using real-world data hereinafter.

## Evaluation of case reports

Four post-marketing cases that relate to suspected insertional mutagenesis were reported to PEI, the national competent authority in Germany, by April 2024 (Table 1). By this time, ~2500 patients had received CAR T therapy in a commercial setting in Germany. These case reports involved three different approved CAR T products. T-cell lymphoma as suspected ADR was reported in three cases. In the fourth case, lymphocytosis with massive CAR expansion but no T-cell malignancy was described. It is noted that T-cell neoplasms can be difficult to diagnose due

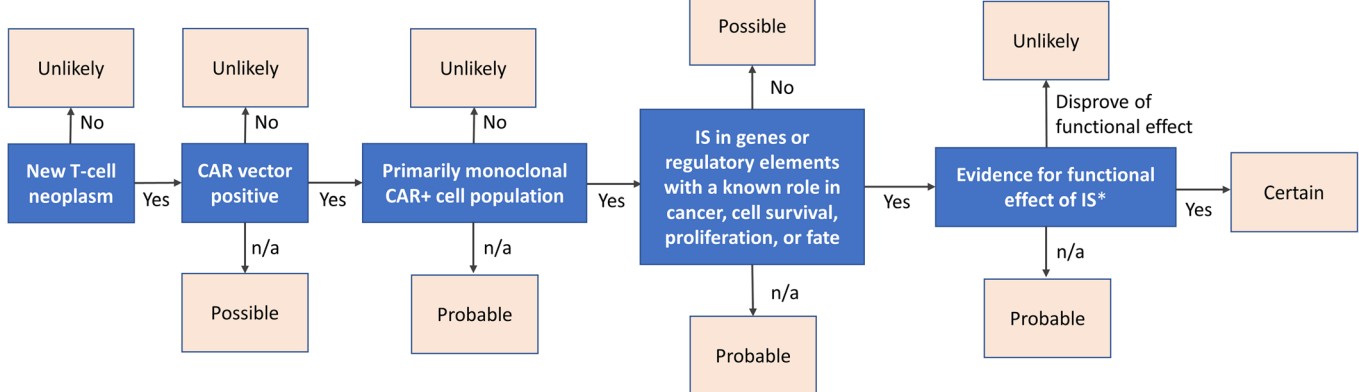

**Figure 1.   Proposed algorithm for causality assessment of suspected insertional mutagenesis triggered T-cell malignancy.**

CAR+ CAR vector positive, IS insertion site(s), n/a not available. *For example: demonstration of differential gene expression or experimental evidence to support IS oncogenic effect.

to their rarity and variability, thus likely resulting in underreporting. Rapid onset with acute toxicities was described in three of the reported cases, two of which included parallel development of haemophagocytic lymphohistiocytosis (HLH) and disseminated intravascular coagulation (DIC). In one case, an onset later than 6 months (following CAR T therapy) with initial skin lesions was described. The available data basis for these reports differed considerably. In case 4 reporting a massive CAR expansion, genetic analysis to address the possibility of insertional mutagenesis was ongoing. However, no T-cell malignancy had been diagnosed. Of the three T-cell lymphomas, case 1 was not tested for the CAR (assessed as possibly related); in case 2 testing was ongoing (assessed as conditional) with several months expected for molecular testing thus delaying the assessment. In case 3, a tumour sample was CAR positive, and a vector integration site analysis indicated a predominant clone with vector integrations into three genomic sites. The insertions were located in the genes *DPF2* (exon), *NPLOC4* (intron), and *RAB11FIP3* (intron) (Kobbe et al, 2024). In line with our criteria for causality evaluation and in the absence of a functional test of the vector integrations (e.g. transcriptome analysis), this case was assessed as probably related to the CAR T therapy. Although the affected genes are not described as oncogenes or tumour suppressor genes, which could have immediately suggested that CAR T therapy contributed to the development of T-cell lymphoma, a role in tumour development cannot be excluded. This

applies also to other genes that have been reported to harbour a vector insertion in two recently published cases of CAR-positive T-cell lymphoma, namely *PBX2* (3'UTR) (Harrison et al, 2023), and *SSU72* (intron) (Ozdemirli et al, 2024). It can be anticipated that whether an integration is oncogenic depends on the precise background in which it occurs. Besides, many cancer driver genes are undiscovered to date (Black and McGranahan, 2021). Notably, several of the mentioned genes harbouring vector integrations in CAR T cells play a role in gene expression regulation: *DPF2* is a regulator of myeloid differentiation (Huber et al, 2017) and was recently proposed as driver gene in T-cell leukaemia/lymphoma (Kogure et al, 2022); *RAB11FIP3* is a neighbouring gene of the tumour suppressor *AXIN1*; PBX2 is a transcription regulator associated with poor prognosis in some cancers (Qiu et al, 2010) and proposed to play a role in gastric tumorigenesis (Zhang et al, 2023); Ssu72 is a phosphatase involved in T-cell differentiation (Lee et al, 2021). Transcriptional profiling of the patients' tumour cells can reveal whether vector insertion influenced the expression of the affected or adjacent genes. This appears to represent a prerequisite for an informed assessment and is required on a case-by-case basis. However, this has only been reported for one CAR-positive T-cell lymphoma so far (Ozdemirli et al, 2024). No such data were available in the cases reported to PEI. An informed causality assessment is currently impaired due to lacking or inconsistent testing and the limited information

reported. Testing might be impeded by the availability of suitable tumour samples, greatly affecting the data basis for the case evaluation. In summary, the causality between the CAR treatment and T-cell malignancy was assessed as possible, probable, conditional, and unlikely for the submitted reports (see Table 1). Differences between CAR T products in reporting rates based on patient exposure are unreliable given the low number of cases reported, but both patient and product-specific factors could play a role as a matter of principle.

## Prominent characteristics of approved CAR T-cell products

To fully understand the authorised CAR T-cell products, it is essential to consider their administration conditions alongside their molecular characteristics. Table 2 highlights the most prominent parameters. All products have turnaround times of 3–4 weeks and are cryopreserved before being administered to patients. Conditioning regimens typically involve cyclophosphamide and fludarabine, with slight variations in dosing and application periods, and start when the drug product has been manufactured (i.e., the CAR T cells are not exposed to the conditioning chemotherapy). The maximum number of CAR T cells administered to adult patients is generally similar, ranging from 1 to $6 \times 10^8$ CAR T cells per patient. For paediatric patients, CAR T-cell doses can be as low as $3 \times 10^6$ CAR T cells. Notable differences between the products include the separation of CD4 and CD8

**Table 1. Characteristics of case reports submitted to PEI relating to suspected insertional mutagenesis after CAR T therapy.**

| Case information | Case 1 | Case 2 | Case 3 | Case 4 |
|---|---|---|---|---|
| Indication | DLBCL | MM | DLBCL, PCNSL | MM |
| **Drug-product characteristics** | | | | |
| Percentage CD3+ cells | 97% | 99% | 99% | 99% |
| CAR-positive T cells | 32% | 13% | 28% | 18% |
| Mean VCN of product (vector/cell) | 0.18 | 0.2* | 0.45 | 0.27* |
| Treatment dose | Not reported | 5.22 × 10e7 | 3.8 × 10e8 | Not reported |
| **Features of adverse reaction report** | | | | |
| Adverse event | TCL | TCL | TCL | Lymphocytosis |
| TTO (in months) | 2 | 10 | 1.5 | <1 |
| Additional toxicities | Sepsis with ARDS | Skin lesions, GI lesions, expansion of CAR-T cells, MM relapse, Wasting Syndrome | HLH with DIC, lymphocytosis | CRS, HLH with DIC, ICANS |
| Outcome | Fatal | Fatal | Fatal | Fatal |
| **Key criteria of tumour sample** | | | | |
| Diagnosis of new T-cell malignancy | Yes | Yes | Yes | No |
| CAR status of sample | Not tested | CAR positive | CAR positive | CAR positive |
| ISA performed | No | In progress | Yes | In progress |
| Sample consist of primarily monoclonal CAR positive cell population | Not tested | Yes | Yes | Unknown |
| VCN of sample (vector/cell) | Not tested | Not tested | 2.36 | Unknown |
| Gene(s) with or adjacent to IS | Not tested | Unknown | DPF2, NPLOC4, RAB11FIP3 | Unknown |
| Gene(s) implicated in cell proliferation, fate, survival, and/or genomic stability? | Not tested | Unknown | Yes | Unknown |
| Differential gene expression tested | Not tested | Not tested | Not tested | Not tested |
| Experimental evidence to support IS oncogenic effect | Not tested | Not tested | Not tested | Not tested |
| **Causality evaluation** | **Possible** | **Conditional** | **Probable** | **Unlikely** |

*ARDS* acute respiratory distress syndrome, *CAR* chimeric antigen receptor, *DLBCL* diffuse large B-cell lymphoma, *GI* gastrointestinal, *IS* vector insertion site, *ISA* insertion site analysis, *MM* multiple myeloma, *PCNSL* primary CNS lymphoma, *TCL* T-cell lymphoma, *TTO* time to onset after CAR T therapy, *VCN* vector copy number.
*To harmonise VCN presentation, VCN reported as copies per transduced cell was transformed by multiplication with the share of CAR-positive cells to indicate copies per cell.

T cells, resulting in a defined ratio of both T-cell subtypes in lisocabtagene maraleucel (Breyanzi). Expansion conditions are optimised for T-cell growth, supported by cytokines such as interleukin (IL)-2, IL-7, and/or IL-15 (Arcangeli et al, 2020; Künkele et al, 2019). While the primary purpose of the expansion period is to increase T-cell numbers, it also eliminates non-T cells by exposing them to culture conditions that are suboptimal to other cell types. This process generates highly enriched T-cell products, even when the manufacturing process starts with mixed peripheral blood mononuclear cells (PBMCs). However, small amounts of natural killer (NK) cells may still be present in the final product as verified for tisagenlecleucel (Kymriah), axicabtagene ciloleucel (Yescarta), idecabtagene vicleucel (Abecma), and ciltacabtagene autoleucel (Carvykti) based on available EMA data (Summary of Product Characteristics [SmPC] or European Public Assessment Report [EPAR]).

Besides molecular characteristics of the CAR, the design of the vectors used for CAR T-cell generation varies (Table 2). While most products are based on LV vectors, two rely on γRV vectors. Of note, LV vectors have been used in the SIN configuration for clinical applications from the beginning on. Importantly, insertional oncogenesis has been regarded as less concerning for T-lymphocytes than for HSCs (Gerdes et al, 2013; Newrzela et al, 2008). Accordingly, brexucabtagene autoleucel (Tecartus) and axicabtagene ciloleucel (Yescarta) received market authorisation despite using conventional γRV vectors without SIN configuration. Beyond differences in vector design, the number of vector copies integrated into the genome of CAR T cells, known as vector copy number (VCN), can significantly impact the likelihood of genotoxicity and oncogenesis. Accordingly, gene therapy guidelines emphasize the need to control VCN to minimise risks (CAT, 2021). In general, it is expected that the target VCN range is justified based on preclinical safety data and early clinical trial results. However, since no specific upper limit for VCN has been established, VCN can vary substantially between products and patients. While there is no public information available about the VCN in the CAR T-cell product of a particular patient (as shown in Table 1), license holders possess

**Table 2.  Characteristics of CAR T products authorised in the EU.**

| Product name (other name) -international nonproprietary name- Year of EU approval | Indication | Target | CAR (scFv-hinge-TMD-coStim-CD3z) | Vector/Env (generation) | Promoter | Turn around time[a] | Starting material | Dosage (CAR +, viable T cells) |
|---|---|---|---|---|---|---|---|---|
| **Kymriah** (CTL-019) -tisagene lecleucel- 2018 | ALL (paediatric), DLBCL, FL | CD19 | FMC63 CD8α CD8α 4-1BB CD3ζ | LV/VSV-G (3rd gen.) | EF1α | 3–4 weeks | Enriched T cells | **ALL:** $0.2–5 \times 10^6$/kg (<50 kg) $0.1–2.5 \times 10^8$ (>50 kg) **NHL:** $0.6–6 \times 10^8$ |
| **Yescarta** (KTE-C19) -axicabtagene ciloleucel- 2018 | DLBCL, PMBCL, FL | CD19 | FMC63 CD28 CD28 CD28 CD3ζ | γRV/GaLV (1st gen.) | MSCV 5′LTR | 2–3 weeks | PBMC | $2 \times 10^6$/kg (max. $2 \times 10^8$) |
| **Tecartus** (KTE-X19) -brexucabtagene autoleucel- 2020 | MCL, ALL (adults) | CD19 | FMC63 CD28 CD28 CD28 CD3ζ | γRV/GaLV (1st gen.) | MSCV 5′LTR | 2–3 weeks | Enriched T cells | **MCL:** $2 \times 10^6$/kg (max. $2 \times 10^8$) **ALL:** $1 \times 10^6$/kg (max. $1 \times 10^8$) |
| **Breyanzi** (JCAR017) -lisocabtagene maraleucel- 2022 | DLBCL, HGBCL, PMBCL, FL | CD19 | FMC63 IgG4 CD28 4-1BB CD3ζ | LV/VSV-G (3rd gen.) | EF1α | 3–4 weeks | CD4 and CD8 T cells, separately | $1 \times 10^8$ (1:1 ratio) |
| **Abecma** (bb2121) -idecabtagene vicleucel- 2021 | MM | BCMA | C11D5.3 CD8α CD8α 4-1BB CD3ζ | LV/VSV-G (3rd gen.) | MND | 4 weeks | PBMC | $4.2 \times 10^8$ |
| **Carvykti** (JNJ-68284528) -ciltacabtagene autoleucel- 2022 | MM | BCMA | VHH1/VHH2 CD8α CD8α 4-1BB CD3ζ | LV/VSV-G (3rd gen.) | EF-1α | 4–5 weeks | Enriched T cells | $0.75 \times 10^6$/kg (max. $1 \times 10^8$) |

*ALL* acute lymphoblastic leukaemia, *BCMA* B-cell maturation antigen, *DLBCL* diffuse large B-cell lymphoma, *Env* envelope protein, *FL* follicular lymphoma, *HGBCL* high-grade B-cell lymphoma, *LV* lentiviral vector, *MCL* mantle cell lymphoma, *MM* multiple myeloma, *NHL* non-Hodgkin lymphoma, *PBMC* peripheral blood mononuclear cells, *PMBCL* primary mediastinal B-cell lymphoma, *γRV* γ-retrovirus vector.
[a]Time period from the moment the patient's cells are received at the manufacturing site until shipped to the health care provider.

this data and could potentially conduct correlation analyses to assess the risk.

## Molecular testing essential for an adequate case evaluation

Insertional mutagenesis is a potential risk for all gene therapies that integrate into the genome. We consider the identification of an insertion site as well as the characterisation of its effect on gene expression highly relevant for the assessment of subsequent malignancies and such analyses have been performed in previous incidents of gene therapy-associated cancer (Hacein-Bey-Abina et al, 2003; Micklethwaite et al, 2021; Goyal et al, 2022; Cesana et al, 2024; Schmidt et al, 2023). Cancer development is a multistep process with increasing genome

instability as underlying characteristic (Hanahan and Weinberg, 2011). Two cases of abnormal, clonal expansion of CAR T have been reported that were attributed to LV insertions in the *TET2* (Fraietta et al, 2018) and the *CBL* gene (Shah et al, 2019), respectively, without a malignant transformation of T cells at the time of reporting. However, such reports underline the potential impact of vector insertions in T cells. Approved CAR T products are authorised for the treatment of relapsed or refractory haematological malignancies and genetic predispositions to develop malignancies present in patients receiving these products may render them more susceptible to insertional mutagenesis. Recapitulating the evolutionary development of a tumour from a single (one time, one tumour location)

sample is difficult (Black and McGranahan, 2021). However, technical advances allow for analysis of neoplasms including T-cell malignancies in increasing detail (Yamagishi et al, 2021) and could help to shed light on the pathogenesis of individual cases in future. Establishing approaches to connect and incorporate multi-patient and multi-site omics data may advance the understanding of insertional mutagenesis, functional consequences, and help to identify potentially predictive factors.

Although the biological mechanism of insertional mutagenesis appears well established, the scientific evaluation of secondary T-cell malignancy cases is impeded by a lack of information with regard to vector presence, clonality, vector integration site(s) as well subsequent analysis of gene expression.

This was true for case reports submitted to PEI as for 38 cases of secondary malignancy of T-cell origin assessed by EMA in 2024, of which only 19 had been tested for the presence of the CAR, revealing seven cases with CAR-positive tumour cells (EMA, 2024). These restrictions limit the understanding of the risk and contributing factors. While the incidence rate of T-cell neoplasms is elevated in the patient population treated with the approved CAR T products (Dores and Morton, 2024), this appears to not fully explain the number of reported secondary malignancies of T-cell origin. Furthermore, the detection of CAR-positive and CAR-negative T-cell malignancies could point to different underlying mechanisms. Some T-cell malignancies might have been triggered by insertional mutagenesis while others could have developed for various reasons. For example, the immune system plays a crucial role in recognising and eliminating incipient cancer cells (Hanahan and Weinberg, 2011) and the risk for cancer is markedly increased in immunocompromised individuals (Vajdic and van Leeuwen, 2009). All patients treated with approved CAR T products receive lymphodepleting chemotherapy prior to treatment and both anti-CD19 and anti-BCMA CARs target B-cells resulting in B-cell depletion, which might cause a temporary loss of premalignant cells being eliminated. In the context of SCID-X1 and WAS, the development of malignancies following gene therapy occurred with a latency of several years, while onset was within weeks or months in the context of CAR T therapy. Additional mechanisms cannot be excluded at this time. Also non-mutational changes may have the potential to affect regulation of gene expression, contribute to clonal expansion and influence cell fate (Hanahan and Weinberg, 2011; Parreno et al, 2024). Effects of the culture of CAR T cells during the manufacturing process which, for example, result in changes to the epigenetic signature (Salz et al, 2023) cannot be fully excluded although there is no clear evidence up to now. The relevance, if any, of different characteristics of approved CAR T products, such as differences in vector type or manufacturing, remains to be established. The development of a CAR-positive T-cell lymphoma was recently described following administration of CD19 CAR T cells generated using a nonviral piggyBac transposon system (Bishop et al, 2021; Micklethwaite et al, 2021). The diagnosis in a first patient led to attentive screening of other patients that had received this therapy, which subsequently resulted in the detection of a T-cell lymphoma in a second patient. While no transposon insertions into typical oncogenes were identified in a detailed investigation, genomic alterations and global changes in gene expression were observed (Micklethwaite et al, 2021). While this approach of CAR T manufacturing differs from the approved products, such findings underline the incomplete understanding of factors relevant for the safety of genetically modified cells and the need for careful surveillance.

## Conclusions

Secondary malignancy of T-cell origin appears to represent a rare adverse reaction after treatment with approved CAR T therapies and necessitates a case-by-case assessment.

The mechanisms contributing to its development may differ depending on the patient and context. Risk factors have not yet been sufficiently identified, and it is currently unclear to which extent factors associated with the product, the conditioning regimen prior to infusion or patient characteristics play a role. Therefore, a better understanding of these factors might allow for risk minimisation in the future.

The informed assessment of pharmacovigilance reports requires a sufficient data basis that is only provided in a minority of cases. Molecular testing of tumour samples for the presence of the CAR vector, the exact location(s) of vector insertion(s), clonality, and effect on gene expression is needed for a sound evaluation. Ideally, this is considered during sample collection (e.g., to allow for transcriptome analysis) and choice of appropriate testing methods (e.g. RNA vs DNA-based methods). More sophisticated analysis might provide further insights.

The evaluation of cases illustrated challenges in the pharmacovigilance of advanced therapy medicinal products. As demonstrated, uniform causality criteria that comply with the characteristics of such therapies are needed to evaluate these adverse reactions. Furthermore, adequate laboratory tests should be used, that provide evidence with regard to insertional mutagenesis.

## Ethics approval

The legal mandate of the Paul-Ehrlich-Institut as a competent national authority covers the risk management of the use of medicinal products and includes the detection, assessment, risk minimisation and communication of adverse reactions.

## Peer review information

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

## Acknowledgements

The authors received no specific funding for this work. The views expressed in this article are the personal views of the authors and may not be understood or quoted as being made on behalf of or reflecting the position of the agencies or organisations with which the authors are affiliated.

## Author contributions

**Philipp Berg**: Conceptualisation; Investigation; Visualisation; Writing—original draft; Writing—review and editing. **Gabriele Ruppert-Seipp**: Conceptualisation; Investigation; Writing—original draft; Writing—review and editing. **Susanne Müller**: Conceptualisation; Investigation; Writing—original draft; Writing—review and editing. **Gabriele D Maurer**: Conceptualisation; Writing—original draft; Writing—review and editing. **Jessica Hartmann**: Conceptualisation; Investigation; Visualisation; Writing—original draft; Writing—review and editing. **Udo Holtick**: Investigation; Writing—original draft; Writing—review and editing. **Christian J Buchholz**: Conceptualisation; Writing—original draft; Writing—review and editing. **Markus B Funk**: Conceptualisation; Supervision; Writing—original draft; Writing—review and editing.

### Funding

## Disclosure and competing interests statement

UH declares honoraria/consultancies from BMS, Janssen, Kite/Gilead, Miltenyi Biotec, and Novartis. The remaining authors declare no competing interests.

