## [Peer Review File · EMBO Molecular Medicine]

CAR T-cell-associated secondary malignancies challenge current pharmacovigilance concepts

Philipp Berg, Gabriele Ruppert-Seipp, Susanne Müller, Gabriele Maurer, Jessica Hartmann, Udo Holtick, Christian Buchholz, and Markus Funk

Corresponding authors: Markus Funk (markus.funk@pei.de) , Christian Buchholz (Christian.Buchholz@pei.de)

Review Timeline:

Submission Date:	11th Oct 24
Editorial Decision:	31st Oct 24
Revision Received:	11th Nov 24
Editorial Decision:	19th Nov 24
Revision Received:	21st Nov 24
Accepted:	22nd Nov 24

Editor: Lise Roth

Transaction Report:

31st Oct 2024

Dear Dr. Funk,

Thank you for submitting your commentary to EMBO Molecular Medicine. I have now received feedback from the reviewers who evaluated your manuscript. As you will see below, both reviewers acknowledge the timeliness and interest of your piece and have only minor suggestions. We will therefore welcome the submission of a revised version of your article that would satisfactorily address the referees' concerns.

We discussed within the team the format that would be most appropriate for your piece, as we think that while it is not a review, a drastic shortening would also be detrimental to the overall message. A possibility would be to make it a Perspective: "A Perspective article should 'set the scene' based on recent developments with an emphasis on future directions of the field of study. Perspectives can take a personal point of view, but should emphasize reported facts and testable hypotheses over speculation and opinion. The format is flexible, but we aim for succinct and focussed discussions to engage a broad readership."

While this format does not yet exist at EMM, it should by the time your paper is accepted. No changes in formatting (length of text or figures/tables) are thus requested. The number of references should be slightly reduced (if feasible without changing too much the main text).

When submitting your revised manuscript, please carefully review the instructions that follow below. We require:

- 1) A .docx formatted version of the manuscript text. Please make sure that the changes are highlighted to be clearly visible.
- 2) Individual production quality figure files as .eps, .tif, .jpg (one file per figure).
- 3) Author contributions: CRediT has replaced the traditional author contributions section because it offers a systematic machine readable author contributions format that allows for more effective research assessment. Please remove the Authors Contributions from the manuscript and use the free text boxes beneath each contributing author's name in our system to add specific details on the author's contribution. More information is available in our guide to authors.
- 4) Disclosure statement and competing interests: We updated our journal's competing interests policy in January 2022 and request authors to consider both actual and perceived competing interests. Please review the policy <https://www.embopress.org/competing-interests> and update your competing interests if necessary.
- 5) As part of the EMBO Publications transparent editorial process initiative (see our Editorial at <http://embomolmed.embopress.org/content/2/9/329>), EMBO Molecular Medicine will publish online a Review Process File (RPF) to accompany accepted manuscripts. This file will be published in conjunction with your paper and will include the anonymous referee reports, your point-by-point response and all pertinent correspondence relating to the manuscript. Let us know whether you agree with the publication of the RPF.

I look forward to receiving your revised manuscript.

Yours sincerely,

Lise Roth
Lise Roth, PhD
Senior Editor
EMBO Molecular Medicine

***** Reviewer's comments *****

Referee #1 (Bridging gap comments for Author):

With more and more CAR T cell therapies approved by the FDA and MEA, CAR T cell-associated secondary malignancies are frequently reported across different institutes, which suggests a systematic evaluation of this issue should be done. Because CAR T cells are so different from traditional treatment modalities, such as small molecules and antibodies, applying conventional assessment to CAR T cells might not be appropriate. The authors summarized the case reports submitted to their institute relating to suspected insertional mutagenesis and gave insightful suggestions to evaluate such risk for patients receiving CAR T treatment. They also discussed several possible mechanisms underlying these secondary malignancies from other CAR T products.

Referee #1 (Remarks for Author):

With more and more CAR T cell therapies approved by the FDA and MEA, CAR T cell-associated secondary malignancies are frequently reported across different institutes, which suggests a systematic evaluation of this issue should be done. Because CAR T cells are so different from traditional treatment modalities, such as small molecules and antibodies, applying conventional assessment to CAR T cells might not be appropriate. The authors summarized the case reports submitted to their institute relating to suspected insertional mutagenesis and gave insightful suggestions to evaluate such risk for patients receiving CAR T treatment. They also discussed several possible mechanisms underlying these secondary malignancies from other CAR T products.

Their experience in handling CAR T cell-associated secondary malignancies is precious, and their recommendations will provide helpful clues in this field. I recommended publishing this paper with minor revisions.

Minor revisions for the authors:

1. To systematically evaluate the insertional mutagenesis, it would be desired to have a database focusing on CAR T cells incorporating different omics data. With larger samples from more patients across different treatment centers receiving CAR T cells, researchers can perform in-depth data mining to explore the potential mechanisms of how this happens. I recommend the authors add this to their manuscript.

2. The tables should be formatted following the traditional method: no vertical line is used to separate the table cells. Some example tables can be found here:

1). Guedan, S. et al. Time 2EVOLVE: predicting efficacy of engineered T-cells - how far is the bench from the bedside? *Journal for immunotherapy of cancer* 10, 1-16 (2022).

2). Cappell, K. M. & Kochenderfer, J. N. A comparison of chimeric antigen receptors containing CD28 versus 4-1BB costimulatory domains. *Nature Reviews Clinical Oncology* 18, 715-727 (2021).

Referee #2 (Bridging gap comments for Author):

The authors highlight not necessarily a gap between bench and bedside, but a more broad discussion on the complexities underlying secondary malignancies in patients treated with CAR T cells. Due to the lack of uniformity among centers and poor tracking/reporting, this is a difficult topic to fully explore. To this end, the authors propose an algorithm that could help streamline the process for identifying causality of these malignancies on a case-by-case basis.

Referee #2 (Remarks for Author):

The submitted manuscript entitled "CAR T-cell-associated secondary malignancies challenge current pharmacovigilance concepts" by Berg et. al discusses the complexities with monitoring and identifying the cause of secondary T cell malignancies in patients received CAR T cell products. This review is timely given the increase in CAR-T patients recently diagnosed with secondary malignancies. It is well written, easy to follow and I like that it provides an algorithm for a more uniform approach to identifying the underlying cause of these diseases. The use of real cases to highlight the complexity of these occurrences was a nice addition. I appreciate the author's work and don't have any changes to request.

Point-to-point Responses

Responses to reviewers' comments are **marked in blue**.

Changes in the manuscript are **marked in red**

Deletions are visualized ~~as that~~.

Reviewers' Comments:

Referee #1 (Remarks for Author):

With more and more CAR T cell therapies approved by the FDA and MEA, CAR T cell-associated secondary malignancies are frequently reported across different institutes, which suggests a systematic evaluation of this issue should be done. Because CAR T cells are so different from traditional treatment modalities, such as small molecules and antibodies, applying conventional assessment to CAR T cells might not be appropriate. The authors summarized the case reports submitted to their institute relating to suspected insertional mutagenesis and gave insightful suggestions to evaluate such risk for patients receiving CAR T treatment. They also discussed several possible mechanisms underlying these secondary malignancies from other CAR T products.

Their experience in handling CAR T cell-associated secondary malignancies is precious, and their recommendations will provide helpful clues in this field. I recommended publishing this paper with minor revisions.

Response: We thank you for reviewing the manuscript and the positive evaluation.

Minor revisions for the authors:

1. To systematically evaluate the insertional mutagenesis, it would be desired to have a database focusing on CAR T cells incorporating different omics data. With larger samples from more patients across different treatment centers receiving CAR T cells, researchers can perform in-depth data mining to explore the potential mechanisms of how this happens. I recommend the authors add this to their manuscript.

Response: We acknowledge the potential benefit of large-scale, multi-omics approaches including participants from different centers in this context.

Change/s in the manuscript: However, technical advances allow for analysis of neoplasms including T-cell malignancies in increasing detail ~~(e.g., [40])~~ and could help to shed light on the pathogenesis of individual cases in future. **Establishing approaches to connect and incorporate multi-patient, multi-site omics data may advance the understanding of insertional mutagenesis, functional consequences, and help to identify potentially predictive factors.**

2. The tables should be formatted following the traditional method: no vertical line is used to separate the table cells. Some example tables can be found here:

1). Guedan, S. et al. Time 2EVOLVE: predicting efficacy of engineered T-cells - how far is the bench from the bedside? Journal for immunotherapy of cancer 10, 1-16 (2022).

2). Cappell, K. M. & Kochenderfer, J. N. A comparison of chimeric antigen receptors containing CD28 versus 4-1BB costimulatory domains. Nature Reviews Clinical Oncology 18, 715-727 (2021):

1. Highlights 1 and 2 remain only loosely supported by the manuscript content

Response: Thank you for highlighting the table formatting. We revised the tables according to the journal guidelines (e.g. no background shading). Since these guidelines state no specifics but refer to the journal style and require the table to be editable, we assume that the final table formatting (e.g., no vertical lines between cells) will take place during typesetting by the journal and no further modifications are required from our side.

Referee #2 (Remarks for Author):

The submitted manuscript entitled "CAR T-cell-associated secondary malignancies challenge current pharmacovigilance concepts" by Berg et. al discusses the complexities with monitoring and identifying the cause of secondary T cell malignancies in patients received CAR T cell products. This review is timely given the increase in CAR-T patients recently diagnosed with secondary malignancies. It is well written, easy to follow and I like that it provides an algorithm for a more uniform approach to identifying the underlying cause of these diseases. The use of real cases to highlight the complexity of these occurrences was a nice addition. I appreciate the author's work and don't have any changes to request.

Response: Thank you; we appreciate the encouraging feedback.

19th Nov 2024

Dear Prof. Funk,

Thank you for submitting your revised manuscript to EMBO Molecular Medicine, and please accept my apologies for the delay in getting back to you as we were in the last steps of setting up the 'Perspective' format in our system.

I appreciate that you addressed the referees' minor concerns. Almost everything is fine now, and I will be able to accept your manuscript once the following editorial issues are addressed:

1. Please rename "Funding" to "Acknowledgements".
2. We can accommodate a maximum of 5 keywords, please adjust accordingly.
3. Please adjust the references format to alphabetical, with 10 authors listed before et al. DOIs should be removed.

Once this is done, please accept all changes.

I look forward to receiving your revised manuscript.

Yours sincerely,

Lise Roth

Lise Roth, PhD

Senior Editor

EMBO Molecular Medicine

The authors addressed the minor formatting issues.

22nd Nov 2024

Dear Prof. Funk,

Thank you for submitting your revised files. I am pleased to inform you that your manuscript is accepted for publication and is now being sent to our publisher to be included in the next available issue of EMBO Molecular Medicine!

Your manuscript will be processed for publication by EMBO Press. It will be copy edited and you will receive page proofs prior to publication. Please note that you will be contacted by Springer Nature Author Services to complete licensing information.

If you have any questions, please do not hesitate to contact the Editorial Office.

Thank you for this interesting contribution to EMBO Molecular Medicine!

With kind regards,

Lise Roth
